# Neurodevelopmental Disorders, Schizophrenia Spectrum Disorders and Catatonia: The “Iron Triangle” Rediscovered in a Case Report

**DOI:** 10.3390/children10010077

**Published:** 2022-12-30

**Authors:** Pamela Fantozzi, Claudia Del Grande, Stefano Berloffa, Greta Tolomei, Carmen Salluce, Antonio Narzisi, Gianluca Salarpi, Barbara Capovani, Gabriele Masi

**Affiliations:** 1IRCCS Stella Maris, Scientific Institute of Child Neurology and Psychiatry, Calambrone, 56018 Pisa, Italy; 2Azienda Usl Toscana Nord-Ovest, Department of Mental Health, Psychiatric Service of Diagnosis and Treatment, Hospital “Santa Chiara”, 38122 Pisa, Italy; 3Department of Clinical and Experimental Medicine, University of Pisa, 56126 Pisa, Italy; 4Azienda Usl Toscana Nord-Ovest, Department of Mental Health, Pontedera, 38122 Pisa, Italy

**Keywords:** catatonia, Autism Spectrum Disorder, Schizophrenia Spectrum Disorder, depression, mania, second-generation antipsychotics, mood stabilizers

## Abstract

Catatonia is a complex neuropsychiatric syndrome, occurring in the context of different psychiatric and neurodevelopmental disorders, in neurological and medical disorders, and after substance abuse or withdrawal. The relationship between Autism Spectrum Disorder (ASD), Schizophrenia Spectrum Disorders (SSDs) and catatonia has been previously discussed, with the three disorders interpreted as different manifestations of the same underlying brain disorder (the “Iron Triangle”). We discuss in this paper the diagnostic, clinical and therapeutic implications of this complex relationship in an adolescent with ASD, who presented an acute psychotic onset with catatonia, associated with mixed mood symptoms. Second-generation antipsychotics were used to manage psychotic, behavioral and affective symptoms, with worsening of the catatonic symptoms. In this clinical condition, antipsychotics may be useful at the lowest dosages, with increases only in the acute phases, especially when benzodiazepines are ineffective. Mood stabilizers with higher GABAergic effects (such as Valproate and Gabapentin) and Lithium salts may be more useful and well tolerated, given the frequent association of depressive and manic symptoms with mixed features.

## 1. Introduction

Catatonia is a complex neuropsychiatric syndrome, firstly described by Kahlbaum in 1874 as a motor syndrome occurring in association with affective disorders, epilepsy and tuberculosis [1]. During the early decades of the twentieth century, Kraepelin and Bleuler included catatonia within the realm of schizophrenia [2,3], contributing to a persistent misclassification over the following 100 years. With the publication of the Diagnostic and Statistical Manual of Mental Disorders, 5th Edition [4], catatonia has been recognized as a condition potentially occurring in the context of different psychiatric disorders (schizophrenia, schizoaffective disorder, bipolar disorder, depression), neurodevelopmental disorders (NDDs), particularly Autism Spectrum Disorder (ASD), in other medical conditions (metabolic, endocrine, rheumatologic, neoplastic and paraneoplastic disorders), in neurological disorders (encephalitis, cerebrovascular diseases, head traumas, etc.), and after substance abuse or withdrawal [5,6,7].

Catatonia is characterized by three specific clusters of motor, speech and behavioral features. DSM-5 diagnostic criteria require three of twelve symptoms, including stupor, catalepsy, waxy flexibility, mutism, negativism, posturing, mannerism, stereotypy, agitation, grimacing, echolalia and echopraxia [4]. However, more than 40 additional manifestations have been described in the literature [5]. Although potentially life-threatening, especially in its malignant form, if catatonia is timely identified, it can respond positively to benzodiazepines (BDZs) as first-line treatment. Electroconvulsive therapy (ECT) is an effective treatment in non-responders to BDZs, malignant catatonia and malignant neuroleptic syndrome [5,8,9,10]. The pathophysiology of catatonia is still unclear, even if several mechanisms have been proposed [11]. Considering the positive response to gamma-aminobutyric acid (GABA) receptor modulators, namely BDZs, the role of the GABAergic system has been identified as a common pathway in the pathogenesis of catatonia [12,13,14]. Traumas, severe stress and grief may also have a triggering effect [15].

Albeit often underdiagnosed or misdiagnosed, catatonia is not rare in pediatric and adolescent populations [13], especially in ASD. Evidence from clinical experience and literature support the notion that catatonia, at least in some of its expressions, is closely linked to NDDs, particularly ASD, often associated with ID [16]. In a systematic examination of 506 individuals with ASD, Wing and Shah [17] found that 17% presented catatonic features. Most of them were males, with symptom onset in the age range of 10–19 years, and with Intellectual Disability (ID) as an associated risk factor. All the NDDs share an early onset, strong persistence over time, frequent co-occurrence, and different developmental pathways [18]. One of these possible pathways is from ASD to the Schizophrenic Spectrum Disorders (SSD) [19,20]. In the NIMH series of children with Very Early Onset Schizophrenia, at least two out of three had premorbid NDDs, and at least one out of four met criteria for ASD prior to the onset of psychotic symptoms [21]. Different studies suggest that ASD patients have a significant higher risk to develop an SSD [22,23,24]. Developmental neuroscience led to the current conceptualization of part of the SSDs as neurodevelopmental disorders starting before birth, presenting a large variety of premorbid and prodromal symptoms leading to progressive impairment for the patient and, in a portion of cases, resulting in the full-blown syndrome [20,25,26]. According to this re-conceptualization of SSDs as expressions of altered neurodevelopmental trajectories, Shorter and Wachtel [27] proposed the concept of “Iron Triangle”, considering the possibility that autism, psychosis and catatonia may be three different manifestations of the same underlying brain disorder.

In this report, we describe the case of a 16-year-old patient with a history of ASD, presenting with an acute-onset severe catatonic symptomatology in the context of an SSD and affective symptoms. Clinical, developmental and treatment implications will be discussed. Particularly, we focus on the treatment implications of this “triad”, in terms of poorer efficacy and tolerability of antipsychotic treatments, with a possible worsening of catatonic symptoms after the increase in antipsychotics dosage.

## 2. Case Presentation

L. is a 16-year-old Caucasian boy with a healthy heterozygous twin and a negative psychiatric family history. He presented in early years isolation and impaired relationships with peers, due to low reciprocity, inflexibility and stereotyped behaviors. These features were associated to a hyperactive and impulsive behavior, inattention with persisting difficulty organizing tasks and activities, and emotional dysregulation, with excessive reactivity and irritability. Intellectual functioning was borderline, with co-occurring learning difficulties and related difficult school achievements since primary school. All these features can be retrospectively referred to a mixed NDD with ASD, ADHD, learning disorders, borderline intellectual functioning and impaired emotional regulation.

During adolescence, isolation, low reciprocity, inflexibility, and stereotyped behaviors and interests persisted, with increasing difficulties with peers. ADHD and emotional dysregulation worsened, with superimposed oppositional and defiant behaviors, hostility, low adherence to social rules, intolerance to frustrations, poor school performances and frequent cannabis use. At age 15, L. and his brother were involved in a fight with other boys, and L. reported a facial trauma with multiple fractures and need for reconstructive surgery. After a few months from this stressful event, and during a phase of cannabis use, L. presented an acute-onset SSD, characterized by persecutory delusions, isolation, disorganized behavior and confusion, agitation (escape from home because of the fear of being killed), and reversal of sleep–wake rhythm. This condition was associated to a severe catatonic symptomatology, according to the Bush-Francis Catatonia Rating Scale [28], a 14-item screening instrument. The patient presented immobility, mutism, negativism, lack of facial expression, staring gaze, poor eye contact and repetitive behaviors. Mixed affective symptoms (depression and guilty feelings, and bizarre disinhibition) were also co-occurring. After a psychiatric assessment, Olanzapine was started at 10 mg/day associated with Lorazepam 3 mg/day. A brain MRI was recommended but not performed due to absence of patient compliance (persecutory delusions). After a partial improvement of psychotic symptoms (but not catatonia), Olanzapine was gradually decreased from 10 to 5 mg/day after 4 weeks of treatment, in the hypothesis of a progressive extinction of the triggering effects of both cannabis and traumatic experience. Catatonia slightly improved, but psychotic symptoms rapidly worsened, with disorganization, persecutory delusion, isolation, increased inflexibility, avoidant attitude towards family members, without response to increased Olanzapine up to 15 mg/day, and, after Olanzapine discontinuation, to Risperidone up to 3 mg/day. After one month, as delusions and disorganized behavior and thoughts were unchanged, as well as flat affect, anhedonia, abulia and social withdrawal, Risperidone was also discontinued, and Lurasidone (74 mg/day, with following increases up to 111 mg/d) and extended-release Lithium salts (12 mmol/day, then up to 24 mmol/L) were introduced. A clinical improvement of speech, thoughts and anhedonia was firstly observed, although moderate catatonia, bizarre ideas, incongruous laughter and social withdrawal were still present, as well as anxiety and sleep/wake rhythm alteration. A switch to Clozapine treatment was proposed, but refused by patient and parents, given the difficulty of taking blood samples. However, a slow and progressive improvement continued in the following months, of both catatonia, psychotic and behavioral symptoms, and mood. At this point, a clinical assessment focused on ASD diagnosis, using the ADOS-2, Module 4 [29] and the ADI-R [30], showed a mild form of ASD, while intellectual functioning was at a borderline level (Total IQ score 73).

After six months, while the patient was still assuming Lurasidone 111 mg/day, a sudden, acute recurrence of illness occurred without apparent triggers (including substance use), with a new, severe worsening of catatonic features (psychomotor slowdown, mutism, negativism, bizarre postures and fixity of gaze), social withdrawal, alternating with agitation, dysphoria, and refusal of medications. At the admittance for hospitalization in emergency room, mutism and negativism, postural stiffness and staring were associated with agitation, irritability, disorganized persecutory thoughts and behavior, inner tension, suspiciousness, episodic elated mood, grandiosity, verbal and physical aggression, episodic inappropriate disinhibition and hypersexual behavior, and decreased need for sleep. The patient was treated with IM Aripiprazole (7.5 mg twice a day) and IM Delorazepam. After a two-week treatment, the agitated catatonia persisted, with the following symptoms: alternation of extreme hyperactivity, unprovoked agitation, impulsivity and hostility, immobility, mutism, negativism, lack of facial expression, staring gaze, poor eye contact, bizarre posture, repetitive behaviors, perseveration and verbiage, refusal to eat and autonomic abnormalities (increased blood pressure and heart rate). On the Bush–Francis Catatonia Rating Scale [28], the patient presented a score of 27 (maximum score 42). During the referral, the following diagnostic assessment was completed: biochemical and hormonal parameters (in the normal range), urine drug screening for substances, including Novel Psychoactive Substances (negative), electroencephalogram (EEG) and contrast-enhanced brain Magnetic Resonance Imaging (MRI) (both without significant abnormalities). The following neuronal antibody profile in blood was examined: NMDAR, GABA_B_, MGA (Myelin-Associated Glycoprotein), anti-neuronal antibodies associated with cerebellar degeneration (HU, YO, RI), and the additional antibody profile for paraneoplastic neurological syndromes (Amphiphysin, CV2, PNMA2 [Ma2/Ta], RI, YO, HU, Revoverin, SOX1, Titin, Zic4, GAD65, Tr DNER).

Given the poor treatment efficacy and the prominence of catatonic and affective symptoms, intravenous (IV) Lorazepam (4 mg vial three times a day) and IV Sodium Valproate (400 mg three times a day), followed after three days by oral Sodium Valproate 1000 mg/day, Lithium salts (up to 20 mmol/L) and Gabapentin 900 mg/day were prescribed, with discontinuation of the antipsychotics. A gradual but evident improvement of catatonic symptoms occurred in the next two weeks. Unfortunately, after the discharge, psychomotor agitation rapidly worsened, with accelerated thought and speech, grandiosity and hypersexuality, so Olanzapine was restarted up to 10 mg/day, with improvement of mood and behavior, but a new worsening of catatonic signs. The antipsychotic was newly reduced to 7.5 mg/day, Lorazepam increased up to 10 mg/day, Lithium salts increased up to 32 mmol/day, Sodium valproate 1000 mg/day and Gabapentin 900 mg/day. Bizarre postures were restricted to parts of the body (prevalently arms), associated to perseveration, suspiciousness, abnormal prosody, bizarre thought and flat mood, and poor eye contact. Psychotic (delusions) and behavioral (hostility) symptoms also moderately improved in the next months, but global functioning has not returned to baseline.

## 3. Discussion

L. presents a complex, multiple neurodevelopmental disorder (NDD), with early impairment of social communication/interaction area, associated with inattention and impulsive/hyperactive symptoms, cognitive impairment (borderline intellectual functioning and learning disorders) and emotion dysregulation, complicated in the secondary school by defiant, oppositional and rule-breaking behaviors. In this neurodevelopmental context of vulnerability, the cannabis use and a significant traumatic event (physical aggression with reconstructive surgery) may have triggered the acute onset of an SSD with both positive and negative symptoms, with prominent component of a catatonic syndrome, co-occurring affective symptoms (both depressive and manic) and a progressive regression in social and adaptive skills. This case report highlights the complex connection between three psychopathological realms (the “Iron triangle”, according to Shorter & Wachtel [27]): (1) ASD, associated with other NDDs in the first years of life, with a developmental course toward a disruptive behavior disorder, emotional dysregulation and a possible triggering role of substance use and traumatic experiences; (2) an acute onset SSD with prominent affective symptoms; (3) a full-blown catatonic syndrome, with waxing and waning course, worsened by antipsychotic medications.

The ASD, with altered peer relationships and low reciprocity, can be a precursor of a schizotypal personality disorder, and possibly of an SSD [22,23]. The link between ASD and catatonia has been repeatedly reported [16,17]. Catatonia, psychosis and ASD have long been considered competing diagnoses or, more recently, entities belonging to a wide spectrum, flip sides of the same coin [27,31,32]. Cannabis use and trauma could represent triggering factors for an acute onset of an SSD. A meta-analysis on child and adolescent psychosis [33] reported a frequent comorbidity with posttraumatic stress disorder (34.3%), ADHD and/or disruptive behavior disorders (33.5%), and substance abuse/dependence (32.0%). Substance use is common in adolescent-onset SCZ, and it strongly interacts with genetic and early epigenetic factors [33].

These intertwined developmental pathways underline the need for a developmental approach to psychopathology, more centered on a careful exploration of the trajectories over the years [34], and of the role of triggering factors [20]. A correct exploration of these developmental pathways may enable us to disentangle a heterogeneous clinical condition (i.e., schizophrenia) in specific subtypes, not only in terms of specific phenotype (i.e., with associated catatonia and/or mixed mood symptoms), but also in terms of treatment specificities [20].

Antipsychotics have been often necessary in our clinical case to manage psychotic symptoms (delusions), behavioral symptoms (impulsivity and aggression), and manic symptoms (elated mood, disinhibition), but their use had negative implications on catatonia, flat affect, anhedonia and abulia. The atypical response to antipsychotic treatments is in line with literature data, suggesting that individuals with comorbid ASD and psychosis may be less likely to experience benefit from multiple antipsychotics [35]. At the same time, patients with catatonia may be at risk for developing neuroleptic malignant syndrome when treated with antipsychotic agents [36,37]. Clozapine, the most effective drug in children and adolescents with SCZ [38,39], may be helpful to reduce psychotic symptoms and improve functional outcomes in patients with psychosis and ASD [40] (but it was not started in our patient because of the patient and family refusal to serial blood samples).

In summary, we submit that a complex clinical condition can be characterized, with ASD (particularly with ID or Borderline Intellectual Functioning and emotional dysregulation), SSD (with mixed schizophrenic and affective symptoms) and catatonia. In this condition, antipsychotics should be used at the lowest dosages, with increases only in the acute phases, when BDZs are poorly effective. The use of mood stabilizers, both those with higher GABAergic effect (such as Valproate and Gabapentin) and Lithium salts, may be more useful and well tolerated.

## Data Availability

The data presented in this study are available on request from the corresponding author.

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
