# Peer review of "Neurodevelopmental Disorders, Schizophrenia Spectrum Disorders and Catatonia: The “Iron Triangle” Rediscovered in a Case Report"

_children, 2022, doi:10.3390/children10010077_

Round 1
Reviewer 1 Report
Point 1. In line 59-61 it says "Evidences from clinical experience and literature support the notion that catatonia, at least in some of its expressions, is closely linked to the NDDs, namely ASD, often associated with ID and Attention-Deficit/Hyper- 61 activity Disorder (ADHD)". I think a reference is needed if you claim that this finding also comes from literature.
Point 2. I suggest asking for the permission of Shorter E and Wachtel L.E. to use the term "Iron triangle", especially if you are using it in the title of your article.
Point 3. In lines 115-116, it is not clear whether Olanzapine and Risperidone were discontinued when Lurasidone and Lithium salts were introduced.
Point 4. Throughout the article, you used the terms "moderate catatonia" and "severe catatonia" (line 77, line 102, line 117). Have you used a scale to measure the severity of catatonia every time you described the symptoms as moderate or severe? If not, I think you need to rephrase, as these levels of severity are not very well defined in DSM 5. Perhaps it would be better to describe the immobility as moderate or severe, or to avoid using these levels of severity altogether.
Point 5. In line 106, you mentioned that "A brain MRI was recommended, but not performed due to absence of patient compliance" and then, in lines 160-161, that "contrast-enhanced Magnetic Resonance Imaging (MRI) of the brain excluded significant abnormalities.". Please clearly specify in the text when was the MRI performed.
Point 6. In line 158, you mentioned that serum auto-antibodies were normal. Please specify which auto-antibodies are you referring to and what types of autoimmune encephalitis you excluded. Did you perform a CSF analysis?
Point 7. There is a significant number of errors in English, I suggest having your article revised by a fluent English speaker.
Reviewer 2 Report
This is a well written and interesting case report article on “ Neurodevelopmental disorders, schizophrenia spectrum disorders and catatonia: The “Iron Triangle” rediscovered in a case report ”. Despite the fact that the main text was written with the preservation of scientific manner and in respect to the journal guidelines, however the reviewer noticed some shortcomings of the manuscript.
-
The list of references in its current form does not meet the requirements of the journal. Revision is mandatory.
-
In the opinion of the reviewer, this manuscript could represent a more scientific expression if the manuscript included the data presented in this paper as a supplement to the reading of the text, rather than simply making such data available upon request addressed directly to the corresponding author.
